# Mechanisms Underlying Connexin Hemichannel Activation in Disease

**DOI:** 10.3390/ijms22073503

**Published:** 2021-03-28

**Authors:** Raf Van Campenhout, Ana Rita Gomes, Timo W.M. De Groof, Serge Muyldermans, Nick Devoogdt, Mathieu Vinken

**Affiliations:** 1Department of Pharmaceutical and Pharmacological Sciences, Vrije Universiteit Brussel, Laarbeeklaan 103, 1090 Brussels, Belgium; Raf.Van.Campenhout@vub.be (R.V.C.); Ana.Rita.Coelho.Gomes@vub.be (A.R.G.); 2Department of Medical Imaging, In Vivo Cellular and Molecular Imaging Laboratory, Vrije Universiteit Brussel, Laarbeeklaan 103, 1090 Brussels, Belgium; Timo.De.Groof@vub.be (T.W.M.D.G.); ndevoogd@vub.be (N.D.); 3Laboratory of Cellular and Molecular Immunology, Vrije Universiteit Brussel, Pleinlaan 2, 1050 Brussels, Belgium; Serge.Muyldermans@vub.be

**Keywords:** connexin hemichannel, pathology, mechanism

## Abstract

Gap junctions and connexin hemichannels mediate intercellular and extracellular communication, respectively. While gap junctions are seen as the “good guys” by controlling homeostasis, connexin hemichannels are considered as the “bad guys”, as their activation is associated with the onset and dissemination of disease. Open connexin hemichannels indeed mediate the transport of messengers between the cytosol and extracellular environment and, by doing so, fuel inflammation and cell death in a plethora of diseases. The present mini-review discusses the mechanisms involved in the activation of connexin hemichannels during pathology.

## 1. Introduction

Although the cell plasma membrane features an impermeable double phospholipid layer structure, cross-membrane trafficking of biomolecules and ions is essential to control various cell processes [1]. Cell plasma membrane transport proteins include different membrane proteins that enable such transfer. Among those are gap junctions, which form cell-to-cell junctions that facilitate direct intercellular communication between cells by allowing the passage of small and hydrophilic molecules, including glucose, glutamate, glutathione, adenosine triphosphate (ATP), cyclic adenosine monophosphate, inositol triphosphate and ions, such as calcium, sodium and potassium [1]. These communicating cell-to-cell junctions serve as gatekeepers for many physiological processes [2,3]. They arise from the interaction of two hemichannels, which in turn are built up by six connexin proteins at the cell plasma membrane surface of adjacent cells. Today, more than 20 different connexin species have been identified. The connexin family members share a common structure consisting of four transmembrane domains, two extracellular loops, one cytosolic loop, one cytosolic carboxyterminal tail and one cytosolic amino tail (Figure 1). The different connexin family members are designated based upon their molecular weight as predicted by cDNA sequencing. In this respect, connexin43 (Cx43), which is the most abundantly expressed connexin variant, has a molecular mass of 43 kDa [1,2,3].

Over the past two decades, it has become clear that connexin hemichannels also provide an autonomous communication pathway for communication on their own, independent of their role as structural precursors of gap junctions [4]. While gap junctions mediate intercellular communication, connexin hemichannels support the transport of messengers between the cytosol and the extracellular environment. Unlike gap junctions, connexin hemichannels become predominantly active under pathological conditions (Table 1) [2,3,5]. The current mini-review provides an overview of the machinery involved in connexin hemichannel opening in disease.

## 2. Role of Connexin Hemichannels in Inflammation and Cell Death

### 2.1. Inflammation

The involvement of connexin hemichannels in inflammation and cell death has been well documented in a variety of studies using genetic knock-out animals, knockdown of connexin expression and connexin hemichannels inhibitors [26,27,28]. Both processes result from tissue responses against infections, chemical insults and physical injury [29,30]. Inflammation is a cohesion of reaction mechanisms that initiates the process of pathogen clearance and tissue repair. At the cellular level, canonical inflammasome activation requires two signals (Figure 2). The interaction of damage-associated and pathogen-associated molecular patterns with Toll-like receptors of immune cells induces inflammation [31]. This promotes the transfer of nuclear factor (NF)-κβ to the nucleus to activate gene expression. Consequently, the genes encoding the premature forms of interleukin (IL)-1β and IL-18 are transcriptionally activated, and the proteins are cleaved to their mature form by caspase 1 in the cytosol. The latter requires a second signal, as nucleotide-binding oligomerization domain leucine rich repeat and pyrin domain-containing protein 3 (NLP3) inflammasome formation leads to activation of caspase 1. Connexin hemichannels play a prominent role in the initiation of inflammation, because they act as activators of the NLP3 inflammasome pathway by releasing ATP [31,32]. Pathogenic stimuli drive the opening of connexin hemichannels, whereby extracellular ATP molecules can stimulate P2X7 receptors. The activation of these P2X7 receptors results in a decrease of intracellular potassium ions, which is a trigger of NLP3 inflammasome activation [33]. In this way, purinergic P2X7 receptors promote activation of the inflammasome pathway to release IL-1β and IL-18 in the extracellular environment and influence the production of other inflammatory mediators, such as IL-6, tumour necrosis factor (TNF)-α and nitric oxide [31,32,34]. This interplay between connexin hemichannels and NLP3 inflammasome activation plays a pivotal role in initiation of disease pathology. In this respect, it has been well documented that aberrant Cx43 hemichannel activity underlies renal damage in chronic kidney disease. Biopsy material from patients with diabetic nephropathy, an inflammatory-associated disease that represents approximately half of patients with end-stage kidney failure, show an upregulation of Cx43 protein production in the tubular epithelia. This increased expression of Cx43 mediates the onset of this disease by mediating the extracellular release of ATP molecules. Thus, P2X7 signalling and NLP3 inflammasome activation are stimulated. In this way, Cx43 hemichannel activity provokes inflammatory damage and phenotypic changes that predispose tubular injury in chronic kidney disease [35,36]. Similarly, activation of the NLP3 inflammasome pathway contributes to diabetic retinopathy, a complication of diabetes that can results in vision loss. By blocking Cx43 hemichannels in an ex vivo human organotypic retinal culture model of diabetic retinopathy, it was seen that NLP3 inflammasome activation was hindered. Consequently, Cx43 hemichannel modulation turns down the release of pro-inflammatory cytokines [37]. In addition, other triggers of NLP3 inflammasome activation, including, pH fluctuation, oxidative stress and calcium ion mobilization, are well-known regulators of connexin hemichannels [38,39].

The inflammatory environment on its own is another triggering factor for connexin signalling and can contribute to disease progression. As such, pro-inflammatory cytokines act as activators of Cx43 hemichannels of mouse astrocytes. While pro-inflammatory treatment reduces gap junction-mediated intercellular communication, a mixture of IL-1β and TNF-α increases Cx43 hemichannel activity in murine astrocytes. This induced opening of Cx43 hemichannels affects the trafficking of glucose molecules. Thus, activated Cx43 hemichannels enhance the cellular uptake of glucose, which might explain the metabolic changes of astrocytes involved in brain inflammation [40]. Similarly, promoted Cx43 hemichannel activity is seen after treating human endothelial cells with IL-1β and TNF-α in combination with glucose, a condition that occurs in cardiovascular diseases [41]. The opening of Cx43 hemichannels is associated with extracellular ATP release, which stimulates mitogen-activated protein kinases, nitric oxide production, cyclo-oxygenase-2 and purinergic and prostaglandin receptors. Thus, several signalling cascades are triggered, leading to endothelial dysfunction and cell damage [41]. The involvement of connexin channels in inflammation is also demonstrated by the differential effects of lipopolysaccharide, the major component of the outer membrane of Gram-negative bacteria, on gap junctions and connexin hemichannels. Lipopolysaccharide induces an inflammatory response by activating the arachidonic acid pathway. This arachidonic acid pathway regulates inflammatory responses through forcing the biosynthesis of prostaglandins and thromboxane A2 from arachidonic acid. While lipopolysaccharide inhibits gap junctions, the response of connexin hemichannels depends on the balance between kinase-mediated phosphorylation of connexins and arachidonic acid effects [42]. In this respect, predominance of the arachidonic acid effect can support pathogenic-pore behaviour by stimulating paracrine ATP signalling [42]. For readers who want to learn more about the details and specific aspects on the involvement of connexin hemichannels in inflammation and associated diseases, references to papers that extensively describe and discuss the prominent role of connexin hemichannels in inflammation have been provided [33,34,43]. In short, studies show that the passage of calcium ions and ATP through connexin hemichannels stimulates inflammatory signalling pathways in different acute and chronic diseases, like acetaminophen-induced liver failure, lung inflammation and diabetic retinopathy [33,34,43]. In this respect, it has become clear that connexin hemichannels mediate cellular communication underlying inflammatory diseases in several organs [33,34,43].

### 2.2. Cell Death

Connexin hemichannels can participate in cell death in many ways. The opening of connexin hemichannels has been observed in different types of cell death, including apoptosis, necrosis, necroptosis and ferroptosis [44,45,46,47]. As such, connexin hemichannels participate in the cellular release and uptake of essential metabolites and toxic substances. Thus, Cx32 hemichannels are involved in neurotoxicity. Mouse microglia, treated with TNF-α, are more prone to cell death as the induced opening of Cx32 hemichannels causes the release of glutamate, which promotes neurotoxicity [48]. Vice versa, cell death also affects connexin hemichannels. Human lens epithelial cells feature functional connexin hemichannels at the cell plasma membrane by expressing Cx32, Cx46 and Cx50 proteins. Exposure of these cells to linoleic acid results in an induction of cell death. Connexin hemichannels are involved in initiation of apoptosis, since treatment of lens cells with linoleic acid opens connexin hemichannels. This leads to elevations in intracellular calcium ion concentrations [49]. Such overload of calcium ions can contribute to cell death via signalling cascades leading to phagocytosis, endoplasmic reticulum stress, mitochondrial permeabilization and nuclear changes. Influx of calcium ions also affects connexin hemichannels and associated transfer of vital molecules [49,50]. Another role of connexin hemichannels in cell death includes communication of messages towards surrounding cells in a paracrine manner [51,52]. Rat glioma cells transfected with *cx43* can be triggered with cytochrome C to undergo apoptosis. Comparison with non-transfected cells demonstrated that both gap junctions and Cx43 hemichannels contribute to the spatial spreading of apoptosis through calcium ion fluxes. While gap junctions can only mediate apoptotic cell death in close proximity, Cx43 hemichannels also affect healthy cells beyond the ‘gap junction-associated area’ [51]. This bystander signalling effect of Cx43 hemichannels has also been shown in brain microvascular endothelial cells that were isolated from mice. X-rays can cause DNA damage and cell death in surrounding cells, and connexin hemichannels are associated with these radiation-induced bystander effects. The opening of Cx43 hemichannels propagates damage to non-irradiated cells by participating in signalling cascades involving calcium ions, reactive oxygen species (ROS), ATP and nitric oxide [52].

## 3. Regulation of Connexin Hemichannels

### 3.1. Mechanical Stimulation

Evidence for the opening of connexin channels in response to mechanical stimulation has been predominantly shown in chicken and murine osteocytes. Osteocytes, the most abundant cells present in skeletal adult bone tissue, are regulators of bone remodelling processes. Bone remodelling is involved in the reshaping and replacement of bone following injury, including fractures. An imbalance of bone remodelling processes results in major bone loss and osteoporosis in patients. Osteocytes play a central role in the initiation of bone remodelling, as they are mechanosensitive cells that sense stress within the bone [53,54]. Mechanical stimulation of bone induces fluid flow in the lacuna canalicular network and osteocytes respond to this shear stress by releasing intracellular prostaglandin E2 (PGE2) via Cx43 hemichannels [6]. Cx43 hemichannel activity in response to mechanical stimulation in osteocytes is adaptive. The opening of Cx43 hemichannels is correlated with the magnitude of fluid flow shear stress [55]. Fluid flow shear stress initiates interaction between integrin α-5, a cell plasma membrane protein, and the carboxyterminal tail of Cx43 [7]. The interplay between these 2 proteins is enhanced by protein kinase B-mediated phosphorylation of Cx43 on serine373. This modification stabilizes complex formation with 14-3-3θ, an adapter protein that regulates Cx43 hemichannel activity by stimulating translocation towards the cell plasma membrane surface [56]. By doing so, the opening of Cx43 hemichannels is enhanced. However, continuous shear stress leads to a gradual closing of Cx43 hemichannels [55]. It has been shown that closure of Cx43 hemichannels is regulated by one of its substrates. The release of PGE2 initiates an accumulation effect that leads to closure of hemichannels by promoting Cx43 phosphorylation through extracellular signal-regulated kinases [8]. Thus, the mechanical stimulation of osteocytes results in open and closed Cx43 hemichannels, and controls extracellular PGE2 levels. PGE2 does not only regulate Cx43 hemichannel activity, but also acts as mediator for the prevention of bone related diseases. PGE2 preserves osteocyte viability, inhibits osteoclast functionality and stimulates differentiation of osteoblasts to increase bone formation [57]. However, the prominent role of PGE2 in bone pathology remains difficult to unravel, as it stimulates osteoclast formation at high concentrations as well [58]. This biphasic effect seems to be important in health and disease. Endogenous levels of prostaglandins regulate bone physiology whereas abnormalities in PGE2 quantities are linked with pathology [59]. Consequently, the connection between Cx43 hemichannel activity through mechanical stimulation and release of PGE2 by osteocytes may be of paramount importance in bone pathology.

### 3.2. pH Fluctuation

The keratitis-ichthyosis-deafness (KID) syndrome is a rare disorder characterized by skin lesions, hearing loss and vascularizing keratitis. KID is associated with mutations in the *cx26* gene that lead to excessive opening of Cx26 hemichannels [18]. One of the Cx26 KID mutations substitutes a valine for alanine at amino acid position 40 (alanine40valine) and affects activity of Cx26 hemichannels. Xenopus oocytes expressing human Cx26 proteins show sensitivity towards pH fluctuation, as adjusting extracellular pH values from 8.0 to 6.5 decreases Cx26 hemichannel currents. Furthermore, this inhibitory effect on Cx26 hemichannels by pH is less recorded with oocytes harbouring the alanine40valine mutant [19]. Given that physiological pH levels reduce Cx26 hemichannel activity and Cx26 protein levels control dermal homeostasis by regulating keratinocyte differentiation and proliferation, it is hypothesized that insensitivity of Cx26 hemichannels towards pH might underlie skin disorders in KID patients [20,21,22]. While the epidermis is slightly acidic, the environment of the deeper layers of the skin is more neutral. Since the alanine40valine point mutation at the boundary of the first transmembrane domain and the first extracellular domain of Cx26 results in abnormal opening of Cx26 hemichannels in this pH range, aberrant Cx26 hemichannel activity of keratinocytes in the basal layers might contribute to the development of hyperkeratosis [19]. Open Cx26 hemichannels give rise to leakage of essential biomolecules, like ATP, and the resulting ionic imbalance mediated by these leaky Cx26 hemichannels is associated with abnormal proliferation of keratinocytes and pathological features [19,21,22]. However, the mechanism of pH-mediated Cx26 hemichannel activity remains elusive. Protonation of Cx26 could promote activation of its hemichannels, since an acidic pH drives a conformational change of Cx26 proteins to reform channels from a closed to an open transition state [23]. However, increasing the extracellular pH can also trigger hemichannel opening. In this respect, human cervical cancer cells transfected with *cx43* show sensitivity towards alkalinization. Raising extracellular pH values from 7.4 to 8.5 leads to an enhanced Cx43 hemichannel activity, which was measured through ethidium uptake, compared to wild-type cervical cancer cells [60].

### 3.3. Calcium Concentration

Another regulator of connexin hemichannel activity is the amount of calcium ions present in the extracellular environment and cytosol. Physiological conditions preserve sufficient high concentrations of surrounding calcium ions to keep connexin hemichannels in a closed state [61]. For hemichannels composed of Cx26, Cx32 and Cx43, it has been shown that reducing the calcium concentration at the extracellular side leads to activation. As such, the cascading decrease of calcium concentration is accompanied by a gradual enlargement of the pore diameter [14,61,62]. This reversible process is explained by the ability of calcium ions to bind extracellular parts of the connexin hemichannel to unbalance the open hemichannel configuration. So, the removal of calcium ions allows rearrangement of connexin proteins, which induces connexin hemichannel-mediated diffusion processes [61]. The interaction between calcium ions and Cx32 hemichannels is mediated by aspartic amino acids present in the second extracellular loop. Two aspartic amino acids of each Cx26 protein are responsible for the formation of a ring of 12 aspartic amino acids in the hemichannel configuration, which is an extracellular region for docking of surrounding calcium ions. In this respect, mutants harbouring site-directed point mutations of human Cx32 at aspartic acid169 or aspartic acid178 in oocytes from *Xenopus laevis* generate hemichannels that are less sensitive towards calcium-ion dependent triggering compared to wild-type hemichannels [14]. This aberrant connexin hemichannel activity is also linked with pathological conditions. The naturally occurring mutation in the *cx32* gene replaces aspartic acid for tyrosine at position 178. This amino acid substitution is known as one of the 260 different mutations causing X-linked Charcot-Marie-Tooth disease, an inherited form of demyelinating neuropathy. Since this mutation is associated with calcium ion dysregulation, uncontrolled opening of hemichannels can induce disturbance of transportation of ions and small molecules across the cell plasma membrane of Schwann cells and underlies the pathogenesis of neuropathy [14,15]. Furthermore, intracellular calcium ions are involved in connexin hemichannel opening as well. As a consequence of larger cytosolic calcium ion concentrations, a calmodulin-depending cascade is activated to open connexin hemichannels via intermediate signalling steps [63,64]. The implication of Cx43 hemichannels and intracellular calcium ions in pathological conditions has been demonstrated by injecting mice with TNF-α to induce a systemic inflammatory response syndrome characterized by overproduction and secretion of cytokines and chemokines into the circulation. Whereas blocking Cx43 hemichannels protects mice against TNF-α-induced mortality, hypothermia and vascular permeability alterations, stimulation of hemichannel opening has the opposite effect. Whole-cell voltage clamp experiments on human cervical cancer cells overexpressing Cx43 show that TNF-α induces Cx43 hemichannel opening depending on calcium [25]. The interplay between intracellular calcium ions and connexin hemichannels is also associated with several neuroinflammatory conditions. As such, open Cx43 hemichannels contribute to degranulation processes in mast cells in Alzheimer’s disease, amyotrophic lateral sclerosis and harmful stress conditions [24].

### 3.4. Changes in Transmembrane Voltage

The voltage regulation of connexin hemichannel activity is mainly a consequence of voltage-driven changes in connexin protein conformation. Differences in electrical potential between the cytoplasmic and extracellular environment change the position of amino acids, whereby gating properties are affected [65]. The voltage-mediated activity of connexin hemichannels shows 2 distinct forms of hemichannel gating, namely a fast and a slow gating mechanism, which are associated with transitions between open and sub-conductance states, and transitions between the open and closed state through intermediate conductance states, respectively [65,66]. This sensitivity towards transmembrane voltage is underscored by voltage sensors that are present in connexin proteins. As such, the slow gating mechanism induces conformational changes of connexin proteins by coordinating a rotation of the first transmembrane domain and a tilt of connexin subunits [66,67]. In this respect, the asparagine amino acid at position 159 of Cx26 proteins is responsible for forming voltage-activated hemichannels. Electrophysiology measurements in xenopus oocytes and murine neuroblastoma cells expressing rat, sheep or human Cx26 proteins show that the single evolutionary amino acid change at position 159 of the rodent protein accounts for voltage insensitive hemichannels. Furthermore, the introduction of a structural change by substituting aspartic acid with asparagine in rat Cx26 restores voltage dependency. Given that mutations within the *cx26* gene affects the functionality of its hemichannels under physiological control of transmembrane voltage, the role of Cx26 mutants and associated hemichannels in pathological activity may be critical [16]. For Cx32, serine85cysteine mutants, which are associated with X-linked Charcot-Marie-Tooth disease, generate more voltage-sensitive connexin hemichannels. Current-voltage relations observed with xenopus oocytes expressing human Cx32 protein and its serine85cysteine mutant show an increased open probability for mutant Cx32 hemichannels present at the cell plasma membrane surface. Since Cx32 is expressed by Schwann cells, the serine85cysteine mutation may cause dysfunction of these cells by abnormalities in trafficking of ions and small metabolites and lead to clinical manifestations of X-linked Charcot-Marie-Tooth disease [17].

### 3.5. Oxidative Stress

Oxidative stress is a critical determinant in the pathogenesis of various diseases, such as ischemia, atherosclerosis and neurodegenerative disorders. The oxidative stress state is characterized by increased ROS formation and impaired antioxidant systems [68,69,70]. When cellular metabolism is stimulated to produce additional ROS, oxidative stress can drive pathological processes by compromising DNA, disrupting membranal layers, inactivating membrane-bound proteins, triggering proteases and affecting signal transduction mechanisms [71]. Cigarette smoke extract and hydrogen peroxide cause opening of connexin hemichannels. Measurements with rat fibroblastoid cells expressing Cx43 and connexin deficient mouse neuroblastoma cells implicate that activation of these hemichannels is evoked by depolarisation of the cell plasma membrane. Oxidative stress and associated hemichannel opening predispose cells to cell death. Following treatment of neuroblastoma cells with cigarette smoke extract, apoptotic changes are observed, while incubation with connexin hemichannel blockers prevents early cell death, cell shrinkage and the formation of apoptotic bodies [72]. In contrast, lens fibre cells benefit from activation of connexin hemichannels. Chicken embryo fibroblast cells transfected with *cx50* are opened by hydrogen peroxide to significantly reduce the level of apoptosis. Counterparts with the Cx50 proline88serine construct, a mutation that is associated with cataract, are rather impeded and no reduction of apoptosis occurs. The protective effect of Cx50 hemichannels against oxidative stress is mediated by the cellular uptake of the antioxidant glutathione [13]. Cx43 hemichannels are also involved in preserving cells against osmotic stress induced-damage. During skeletal aging, which is associated with accumulation of ROS and osteocyte cell death, bones become fragile and more likely to break. The critical role for Cx43 hemichannels in this pathogenesis has been demonstrated by treating murine osteocyte-like cells with hydrogen peroxide. The latter evokes cell death, but a dose-dependent effect on Cx43 hemichannel activity was seen as well. Since blocking of Cx43 hemichannels exacerbates cell death, a protective role for open Cx43 hemichannels has been shown in osteocytes [73].

### 3.6. Phosphorylation

Connexin proteins can undergo several posttranslational modifications, including phosphorylation [74,75]. In fact, phosphorylation is a major regulator of connexin hemichannel opening [76,77]. With the exception of Cx26, all connexins are phosphoproteins. Connexins are substrates for many kinases and the outcome of this posttranslational modification depends on the nature of the kinase and connexin as well on the cellular context [5,75]. In this context, phosphorylation of serine at position 368 of Cx43 controls connexin hemichannel communication. Following injection of *Xenopus laevis* oocytes with Cx43 cRNA, uptake of carboxyfluorescein measurements show that blocking of protein kinase C, a member of the mitogen-activated protein kinase family, increases connexin hemichannel permeability [77]. As expected, the phosphorylation status of Cx43 proteins plays a critical role during disease [9]. Cx43 phosphorylation and dephosphorylation are associated with cardiac ischemia/reperfusion injury. Ischemia leads to dephosphorylation of Cx43 and redistributes gap junction constituents away from intercalated disks. In this way, Cx43 proteins lose their supportive role of coordinating contractile activation and trigger arrythmias in the ischemic heart [10]. Furthermore, relocated Cx43 may operate as open hemichannels. Preventing the loss of critic metabolites, restoring ionic imbalance, hindering cellular swelling and cell rupture via blocking these channels all are protective against myocardial ischemia/reperfusion injury [11,12]. However, it remains difficult to determine a general mechanism for the effects of phosphorylation on connexin hemichannel activity as there is a difference between the connexin family members. Whereas protein kinase C-mediated phosphorylation induces the closure of Cx30 hemichannels in *Xenopus laevis*, Cx43 hemichannels were unaffected. In this regard, the phosphorylation-dependent regulation of connexin hemichannels is seen as a connexin specie-specific process [78]. Furthermore, protein kinase C-isoforms can differentially modulate connexin hemichannel activity as well. Patch clamp analysis with cells, which were derived from human embryonal kidney cells and transfected with *cx43*, shows a difference in Cx43 hemichannel electrical conductance upon treatment with various protein kinase inhibitors [79].

## 4. Conclusions and Perspectives

Connexins have been detected in virtually all cell types and organs. They gather in a hexameric hemichannel configuration at the cell plasma membrane surface. Connexin hemichannels become predominantly active in pathological conditions, and support inflammation and cell death. Hereby, open connexin hemichannels are involved in a wide array of pathologies [2,10,80,81]. Different mechanisms underly connexin hemichannel activity, as reviewed in this manuscript. However, gaining more in-depth insight into the molecular mechanism remains challenging. Since gap junctions and connexin hemichannels are composed of the same connexin building blocks and allow the passage of identical molecules and ions, it is difficult to distinguish between both channel types. Thus far, techniques monitoring extracellular release of messengers, like ATP, or cytosolic uptake of tracer dyes, such as Lucifer Yellow, are widely used to study connexin hemichannel activity, yet the presence of many other cell plasma membrane transport proteins can interfere with these read-outs [82]. Most importantly, this research area lacks robust connexin hemichannel blockers [2]. At present, connexin hemichannel research still largely relies on the use of peptide-based inhibitors that reproduce specific amino acid sequence in the structure of connexin proteins. In this regard, Gap19 is a synthetic nonapeptide (KQIEIKKFK) targeting the cytoplasmic loop of Cx43. Gap19 prevents interaction between the cytoplasmic loop and carboxyterminal tail, leading to inhibition of Cx43 hemichannels, while leaving Cx43-based gap junctions unaffected [83]. Similarly, Gap24 (GHGDPLHLEEVKC) imitates a sequence in the cytoplasmic loop of Cx32 and inhibits its hemichannels [83]. Despite the promising potential of these mimetic peptides both in vitro and in vivo, their short half-life impedes clinical application [84]. Several groups are therefore currently focusing on novel strategies to overcome this issue, including improving stability and selectivity of connexin mimetic peptides [84]. The design of peptides that mimic the carboxyterminal tail of connexin proteins seems particularly interesting for future development of connexin hemichannel inhibitors. Given that amino acid sequences of the carboxyterminal tail differ considerably among connexin species and that this region controls opening of connexin hemichannels, the carboxyterminal tail is considered as a major potential target for peptide-based inhibitors. This is underscored by the αCT1 peptide, which is currently being tested in clinical trials for the treatment of diabetic foot and venous leg ulcers by switching off Cx43 hemichannel signalling [84]. In parallel, antibodies specifically targeting connexin hemichannels are being explored [85]. An additional strategy to develop connexin hemichannel inhibitors includes the modulation of existing drugs. Since it was recently found that aminoglycosides, which are broad-spectrum antibiotics, can act as inhibitors of connexin hemichannels, the idea of using small-molecule drugs for the development of new connexin hemichannel inhibitors is gaining more attention [86,87]. In this respect, the generation of kanamycin derivatives has yet resulted in the identification of a lead compound for Cx43 hemichannel inhibition [88]. Another promising small-molecule that regulates connexin hemichannel activity is tonabersat, a member of the benzoylamino-benzopyran family. Tonabersat is a compound that has been identified as Cx43 hemichannel blocker and shows great potential to impede the role of open connexin hemichannels in inflammation. The drug product of tonabersat, which is currently in clinical trials for the treatment of diabetic retinopathy and age-related macular degeneration, is applicable for oral administration and aims to hinder Cx43-mediated ATP release promoting NLP3 inflammasome activation [89]. Research in these directions should be strongly encouraged, as this may lead to a new generation of drugs to treat a plethora of connexin hemichannel-related pathologies.

## Figures and Tables

**Figure 1 ijms-22-03503-f001:**
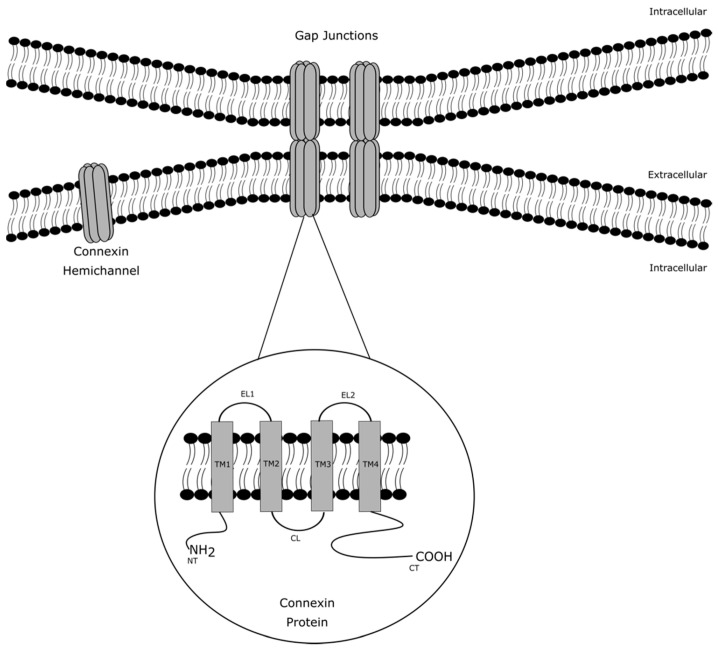
Architecture of gap junction, connexin hemichannels and connexin proteins. Gap junctions arise from the interaction of two connexin hemichannels of adjacent cells. A connexin hemichannel is built up by six connexin proteins. A connexin protein consists of four transmembrane domains (TM1-4), two extracellular loops (EL1-2), one cytosolic loop (CL), one cytosolic carboxyterminal tail (CT) and one cytosolic amino tail (NT).

**Figure 2 ijms-22-03503-f002:**
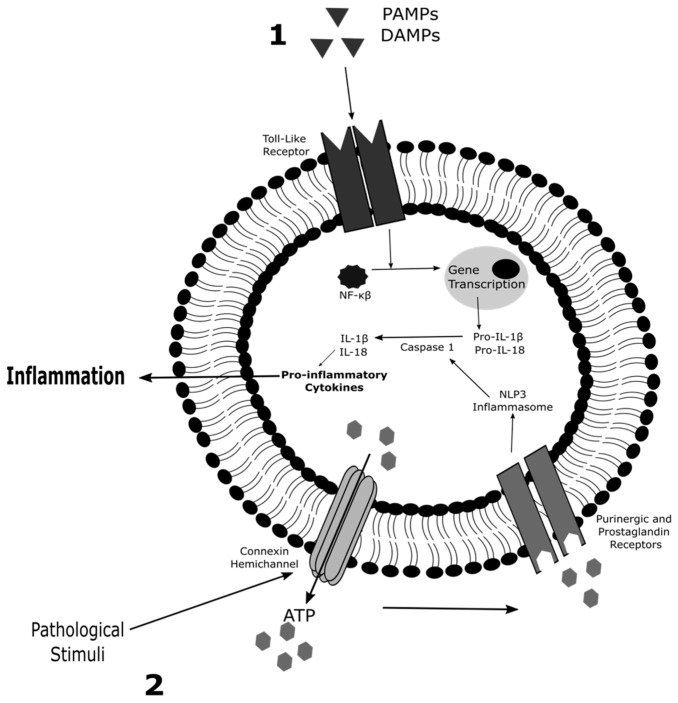
Role of connexin hemichannels in canonical inflammation activation. Canonical inflammasome activation requires two signals. (1) Damage-associated molecular patterns (DAMPs) and pathogen-associated molecular patterns (PAMPs) interact with Toll-like receptors of immune cells to induce inflammation. The binding of DAMPs and PAMPs promotes the transfer of nuclear factor (NF)-κβ to the nucleus to activate gene expression. Thus, the transcription of the genes encoding premature forms of interleukin (IL)-1β and IL-18 is triggered. (2) Pathogenic stimuli drive the opening of connexin hemichannels, promoting the extracellular release of adenosine triphosphate (ATP). Extracellular ATP molecules stimulate P2X7 receptors, leading to nucleotide-binding oligomerization domain leucine rich repeat and pyrin domain-containing protein 3 (NLP3) inflammasome activation. NLP3 inflammasome activation triggers the onset of caspase 1, which influences the inflammatory process by cleaving pro-IL-1β and pro-IL-18 to their mature form and producing other pro-inflammatory cytokines.

**Table 1 ijms-22-03503-t001:** Mechanisms underlying connexin hemichannel activation in disease.

Pathological Condition	Connexin Species	Mechanism of Connexin Hemichannel Activation	References
Bone remodelling processes	Cx43	Mechanical stimulation	[6,7,8]
Cardiac ischemia/reperfusion injury	Cx43	Phosphorylation of connexin proteins	[9,10,11,12]
Cataract	Cx50	Oxidative stress	[13]
Charcot-Marie-Tooth disease	Cx32	Extracellular calcium ion concentration	[14,15]
		Changes in transmembrane voltage	[16,17]
Keratitis-ichthyosis-deafness syndrome	Cx26	pH fluctuation	[18,19,20,21,22,23]
Neuroinflammatory conditions	Cx43	Intracellular calcium ion concentration	[24]
Systemic inflammatory response	Cx43	Intracellular calcium ion concentration	[25]

## Data Availability

Not applicable.

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
