# Peer review of "Mechanisms Underlying Connexin Hemichannel Activation in Disease"

_ijms, 2021, doi:10.3390/ijms22073503_

Round 1

Reviewer 1 Report

  1. Section 2.1 This section is focussed on ATP induced activation of the NLRP3 complex which has been shown to play a pivotal role in initiation of chronic sterile inflammation in multiple tissue types. Given this is a review and not a commentary, I was left wanting more. I would like to see expansion of the work which relates Cx hemichannels to NLRP3 activation and initiation of disease pathology. Are there studies which successfully negate this activity, for example a the back of the eye, where NLRP3 has been deemed a principle downstream initiator of altered Cx43 activity ? The section around NLRP3 priming and activation could also be developed

  1. Line 102. By doing so, pro-inflammatory treatment increases glucose uptake in astrocytes, which underscores metabolic changes of astrocytes involved in brain inflammation. How does activation of Cx43 hemichannels lead to this ? Given this is a review on cx hemichannels and how they mediate damage, the link should be made clear and the evidence detailing the outcome provided

  1. The role of HC in inflammation is well studied but perhaps not truly showcased as it should be in this section. I think the paper would benefit from more evidence around the vast array of studies performed in different models of disease. In addition, greater detail should be provided.

  1. For all sections that follow, I think greater detail can be provided along with additional schematics to highlight relevant pathways. For example the section on phosphorylation is very short and could benefit from expansion

  1. The discussion is fine, but I would like to see inclusion of Cx43 inhibitor Tonabersat which is already in clinical trials for AMD and DR

Author Response

Reviewer 1:

  1. Section 2.1 This section is focussed on ATP induced activation of the NLRP3 complex which has been shown to play a pivotal role in initiation of chronic sterile inflammation in multiple tissue types. Given this is a review and not a commentary, I was left wanting more. I would like to see expansion of the work which relates Cx hemichannels to NLRP3 activation and initiation of disease pathology. Are there studies which successfully negate this activity, for example a the back of the eye, where NLRP3 has been deemed a principle downstream initiator of altered Cx43 activity ? The section around NLRP3 priming and activation could also be developed

This is a very good suggestion. However, the authors want to emphasize that the present manuscript is a mini-review paper, which is now explicitly stated in the abstract, and not a full review paper. For this reason, in-depth discussions have not been included, but rather appropriate references to relevant papers have been provided for readers who want to learn more about specific aspects. Nevertheless, as requested by the reviewer, more statements regarding the interplay between connexin hemichannels and NLP3 inflammasome activation are now included in the manuscript:

“This interplay between connexin hemichannels and NLP3 inflammasome activation plays a pivotal role in initiation of disease. In this respect, it has been well documented that aberrant Cx43 hemichannel activity underlies renal damage in chronic kidney disease. Biopsy material from patients with diabetic nephropathy, an inflammatory-associated disease that represents approximately half of patients with end-stage kidney failure, show an upregulation of Cx43 protein production in the tubular epithelia. This increased expression of Cx43 mediates the onset of this disease by mediating the extracellular release of ATP molecules. Thus, P2X7 signalling and NLP3 inflammasome activation are stimulated. In this way, Cx43 hemichannel activity provokes inflammatory damage and phenotypic changes that predispose tubular injury in chronic kidney disease (Price et al., 2020; Hills et al., 2018). Similarly, activation of the NLP3 inflammasome pathway contributes to diabetic retinopathy, a complication of diabetes that can results in vision loss. By blocking Cx43 hemichannels in an ex vivo human organotypic retinal culture model of diabetic retinopathy, it was seen that NLP3 inflammasome activation was hindered. Consequently, Cx43 hemichannel modulation turns downs the release of pro-inflammatory cytokines (Mugisho et al., 2021).”

  1. Line 102. By doing so, pro-inflammatory treatment increases glucose uptake in astrocytes, which underscores metabolic changes of astrocytes involved in brain inflammation. How does activation of Cx43 hemichannels lead to this ? Given this is a review on cx hemichannels and how they mediate damage, the link should be made clear and the evidence detailing the outcome provided

We agree and have modified the statement to underscore the role of Cx43 hemichannels:

“As such, pro-inflammatory cytokines act as activators of Cx43 hemichannels of mouse astrocytes. While pro-inflammatory treatment reduces gap junction-mediated intercellular communication, a mixture of IL-1b and TNF-a increases Cx43 hemichannel activity in murine astrocytes. This induced opening of Cx43 hemichannels affects the trafficking of glucose molecules. Thus, activated Cx43 hemichannels enhance the cellular uptake of glucose, which might explain the metabolic changes of astrocytes involved in brain inflammation.”

  1. The role of HC in inflammation is well studied but perhaps not truly showcased as it should be in this section. I think the paper would benefit from more evidence around the vast array of studies performed in different models of disease. In addition, greater detail should be provided.

The role of connexin hemichannels in inflammatory diseases is further elaborated in the revised manuscript. Given that this is a mini-review paper, relevant papers have been provided for readers who want to read more on the role of connexins hemichannels in inflammation.

“For readers who want to learn more about the details and specific aspects on the involvement of connexin hemichannels in inflammation and associated diseases, references to papers that extensively describe and discuss the prominent role of connexin hemichannels in inflammation have been provided (Willberords et al., 2016; Sáez and Green, 2018, Zhou et al., 2019). In short, studies show that the passage of calcium ions and ATP through connexin hemichannels stimulates inflammatory signalling pathways in different acute and chronic diseases, like acetaminophen-induced liver failure, lung inflammation and diabetic retinopathy (Willberords et al., 2016; Sáez and Green, 2018; Zhou et al., 2019). In this respect, it has become clear that connexin hemichannels mediate cellular communication underlying inflammatory diseases in several organs (Willberords et al., 2016; Sáez and Green, 2018; Zhou et al., 2019).”

  1. For all sections that follow, I think greater detail can be provided along with additional schematics to highlight relevant pathways. For example the section on phosphorylation is very short and could benefit from expansion

The authors understand the reviewer’s comment, but again we want to emphasize that this manuscript is a mini-review paper. Nonetheless, greater detail was provided by adding more statements. The section on phosphorylation has also more elaborated in the revised manuscript:

“However, increasing the extracellular pH can also trigger hemichannel opening. In this respect, human cervical cancer cells transfected with Cx43 show sensitivity towards alkalinization. Raising extracellular pH values from 7.4 to 8.5 leads to an enhanced Cx43 hemichannel activity, which was measured through ethidium uptake, compared to wild-type cervical cancer cells (Schalper et al., 2010).

“Cx43 hemichannels are also involved in preserving cells against osmotic stress induced-damage. During skeletal aging, which is associated with accumulation of ROS and osteocyte cell death, bones become fragile and more likely to break. The critical role for Cx43 hemichannels in this pathogenesis has been demonstrated by treating murine osteocyte-like cells with hydrogen peroxide. The latter evokes cell death, but a dose-dependent effect on Cx43 hemichannel activity was seen as well. Since blocking of Cx43 hemichannels exacerbates cell death, a protective role for open Cx43 hemichannels has been shown in osteocytes (Kar et al., 2013).”

“However, it remains difficult to determine a general mechanism for the effects of phosphorylation on connexin hemichannel activity as there is a difference between the connexin family members. Whereas protein kinase C-mediated phosphorylation induces the closure of Cx30 hemichannels in Xenopus laevis, Cx43 hemichannels were unaffected. In this regard, the phosphorylation-dependent regulation of connexin hemichannels is seen as a connexin specie-specific process (Alstrøm et al, 2015). Furthermore, protein kinase C-isoforms can differentially modulate connexin hemichannel activity as well. Patch clamp analysis with cells, which were derived from human embryonal kidney cells and transfected with Cx43, shows a difference in Cx43 hemichannel electrical conductance upon treatment with various protein kinase inhibitors (Hawat and Baroudi, 2008).”

  1. The discussion is fine, but I would like to see inclusion of Cx43 inhibitor Tonabersat which is already in clinical trials for AMD and DR

The authors would like to thank the reviewer for this remark. Tonabersat is now included in the conclusions and perspectives section:

“Another promising small-molecule that regulates connexin hemichannel activity is tonabersat, a member of the benzoylamino-benzopyran family. Tonabersat is a compound that has been identified as Cx43 hemichannel blocker and shows great potential to impede the role of open connexin hemichannels in inflammation. The drug product of tonabersat, which is currently in clinical trials for the treatment of diabetic retinopathy and age-related macular degeneration, is applicable for oral administration and aims to hinder Cx43-mediated ATP release promoting NLP3 inflammasome activation (Louie et al., 2021).”

Reviewer 2 Report

This review provides a nice update on the hemichannels that, instead of being viewed as a structural precursors of gap junctions, operate as an autonomous communication pathway in the context of pathological conditions. The authors have done an exhaustive review on the recent publications (most of the cited articles were published in the past 10 years). The table 1 is very informative about the correlations of various hemichannels to diseases. The role of hemichannels in inflammation and cell death were clearly elaborated. The challenges for future study, such as the lack of specific hemichannel blockers to differentiate the pathological contribution of hemichannels vs. gap junctions, were pointed out. Overall, this is a well-written and easy-to-follow review article.

Minor issues:

Line 20: change “manuscript reviews” to “article reviews”

Line 56: please change “manuscript” to “review”.

Author Response

Reviewer 2:

Minor issues:

  1. Line 20: change “manuscript reviews” to “article reviews”

This sentence has been adapted in the revised manuscript to “mini-review discusses”.

“The present article reviews the mechanisms involved in the activation of connexin hemichannels during pathology.”

  1. Line 56: please change “manuscript” to “review”.

This sentence has been adapted in the revised manuscript.

“The current mini-review provides an overview of the machinery involved in connexin hemichannel opening in disease.”

Reviewer 3 Report

The manuscript ‘Mechanisms underlying connexin hemichannel activation in disease’ focuses on the machinery involved in connexin hemichannel opening in disease. The title of this paper is clear and concise, and allows the reader to grasp the context of the manuscript immediately. Van Campenhout et al. start by putting focus on the role of open connexin hemichannels in inflammation and cell death and then provide an overview of triggers that cause connexin hemichannel opening. By doing so, the authors cover an interesting and timely topic in a pressworthy manuscript. The manuscript provides accurate, comprehensive and up-to-date information to readers. The paper summarizes the current understanding of mechanisms underlying connexin hemichannel activation in disease by using professional and scientific language. Furthermore, 2 figures, a table and a graphical abstract, which substantially contribute to the content, were included. This gives the opportunity to the reader to understand the paper through a visual content. However, figure 1 and figure 2 should be slightly adapted. Regarding figure 1, the size of the connexin protein is quite small. Enlarging the connexin protein will be helpful for the reader. Figure 2 can be confusing for readers as a lot of arrows are displayed. It can be recommended to reduce the number of arrows or remove terms like “opening” or “activation” to highlight the role of open connexin hemichannels in figure 2. 

Recommendation: minor revision.

Author Response

Reviewer 3:

  1. Figure 1 and figure 2 should be slightly adapted. Regarding figure 1, the size of the connexin protein is quite small. Enlarging the connexin protein will be helpful for the reader. Figure 2 can be confusing for readers as a lot of arrows are displayed. It can be recommended to reduce the number of arrows or remove terms like “opening” or “activation” to highlight the role of open connexin hemichannels in figure 2.

We modified both figures. Figure 1 and figure 2 were adapted according to the suggestions.

Round 2

Reviewer 1 Report

The authors have taken on board my comments and overall this now represents a nice mini review summarising the field.